# Metagenome-Based Exploration of Bacterial Communities Associated with Cyanobacteria Strains Isolated from Thermal Muds

**DOI:** 10.3390/microorganisms10122337

**Published:** 2022-11-25

**Authors:** Sébastien Halary, Sébastien Duperron, Justine Demay, Charlotte Duval, Sahima Hamlaoui, Bérénice Piquet, Anita Reinhardt, Cécile Bernard, Benjamin Marie

**Affiliations:** 1UMR 7245, CNRS/MNHN, Molécules de Communication et Adaptation des Micro-Organismes (MCAM), équipe “Cyanobactéries, Cyanotoxines et Environnement”, 12 rue Buffon-CP 39, CEDEX 05, 75231 Paris, France; 2Thermes de Balaruc-Les-Bains, 1 rue du Mont Saint-Clair BP 45, 34540 Balaruc-Les-Bains, France; 3Electron Microscopy Platform, Muséum National d’Histoire Naturelle, CP 39, 12 rue Buffon, CEDEX 05, 75231 Paris, France

**Keywords:** cyanobacteria, cyanosphere, heterotroph bacteria, metagenomics, functional redundancy

## Abstract

Cyanobacteria constitute a pioneer colonizer of specific environments for whom settlement in new biotopes precedes the establishment of composite microbial consortia. Some heterotrophic bacteria constitute cyanobacterial partners that are considered as their cyanosphere, being potentially involved in mutualistic relationships through the exchange and recycling of key nutrients and the sharing of common goods. Several non-axenic cyanobacterial strains have been recently isolated, along with their associated cyanospheres, from the thermal mud of Balaruc-les-Bains (France) and the biofilms of the retention basin where they develop. The community structure and relationships among the members of the isolated cyanobacterial strains were characterized using a metagenomic approach combined with taxonomic and microscopic descriptions of the microbial consortia. The results provided insights into the potential role and metabolic capabilities of the microorganisms of thermal mud-associated cyanobacterial biofilms. Thus, the physical proximity, host-specificity, and genetic potential functions advocate for their complementarity between cyanobacteria and their associated microbiota. Besides these findings, our results also highlighted the great influence of the reference protein database chosen for performing functional annotation of the metagenomes from organisms of the cyanosphere and the difficulty of selecting one unique database that appropriately covers both autotroph and heterotroph metabolic specificities.

## 1. Introduction

Cyanobacteria belong to an ancient group of photosynthetic prokaryotes which present a broad range of cellular strategies, physiological capacities, and adaptations that support their occupation of diverse environments worldwide [1]. Cyanobacteria thus constitute pioneer colonizers of specific environments, and their settlement in new biotopes precedes the establishment of intricate microbial consortia [2,3]. Indeed, these autotrophic micro-organisms rarely live in isolated populations but thrive into complex communities where intricate interactions take place [4].

Cultures of isolated cyanobacterial strains typically contain coexisting heterotrophic bacterial partners that are considered as their phycosphere, so-called cyanosphere [5]. Although axenic cultures of cyanobacteria can be achieved, cyanobacterial cultures comprising associated coexisting bacterial populations are more stable and have more robust and longer lifespans than axenic ones [6,7]. Interactions between photoautotroph and heterotroph microorganisms in both natural environments and laboratory conditions are thought to be driven by mutualistic relationships through the exchange and recycling of key nutrients, and sharing of common goods [8]. However, these interaction networks comprise diverse heterotrophic bacteria associated with few photoautotrophic taxa, among which a myriad of inter-species relationships probably rather range from cooperative to competitive, not all being necessarily synergistic. Exploring the structure and role of phototroph-associated microbiota is becoming a topic of major interest, with the potential to reveal relevant emerging properties of these entangled communities, including in the production of molecules of interest [9].

Mud from thermal baths has long been recognized as a healing treatment for arthro-rheumatic diseases, and is colonized by cyanobacterial mats presumed to produce bio-active compounds that can significantly contribute to the health benefits of the cure [10]. Interestingly, in the absence of mud, there is usually no development of a cyanobacterial biofilm in the aquatic substrate despite the addition of nutrients, suggesting that this biofilm occupies a highly specific ecological niche [5]. Cyanobacteria occupy the top layer of said microbial mats, allowing them to harvest light for photosynthesis, and are essentially, but not exclusively, aerobic organisms. Most, if not all, of these cyanobacteria have the ability to secrete an exopolysaccharide sheath around groups of trichomes, forming supracellular rope-like structures called bundles that can attach to mud particles [11]. By stabilizing and structuring this micro-habitat, biofilms can mature and become colonized by a diverse bacterial community constituting a specific cyanosphere [12,13]. Microbial mats retrieved in temperate temperature thermal muds occur in geographically distant locations and mostly include non-heterocytous filamentous cyanobacteria, predominantly *Microcoleus* and *Oscillatoria*, together with small spherical *Chroococcus*-like cyanobacteria and some eukaryotic micro-algae (mainly Diatomophyceae) [13,14,15,16]. Still, little is known about the heterotrophic bacteria associated with cyanobacterial mats from thermal muds, despite that this biofilm-forming microbial community is distinguished by high primary productivity and high rates of N_2_ fixation [17,18].

Recently, studies have investigated the structure of cyanosphere communities using omics-based methods such as metagenomics, and have suggested the occurrence of intertwined metabolic pathways between cyanobacteria such as *Microcystis* and surrounding bacteria [19]. Few studies have yet provided metagenomics-based analyses of the cyanosphere in freshwater ecosystems, and observed a high complementarity between the metabolic potential of the cyanobacterial and the surrounding bacterial partners [20].

In the present study, we aimed to characterize the community structure and relationships among members of a cyanobacterial biofilm, using a metagenomic approach combined with taxonomic and microscopic description of the microbial consortia constituting the cyanosphere. Nine non-axenic cyanobacterial strains were used, recently isolated with their associated cyanosphere from biofilms occurring in the retention basin of the thermal mud of Balaruc-les-Bains (France) [16]. The results provided insights into the potential role and metabolic interaction of bacteria of thermal mud-associated cyanobacterial biofilms.

## 2. Materials and Methods

### 2.1. Field Sampling and Cyanobacteria Isolation

Mud and biofilm samples were collected in 2014 between 28 April and 13 October twice monthly (Figure 1) from the mud maturation basin Balaruc-Les-Bains’s Thermes (43°26′44.0″ N; 3°40′29.6″ E) for phytobenthic community analysis and cyanobacterial strain isolation (Figure 1), as previously described [16].

The retention basin (500 m^3^) maintains a water level above the mud and the continuous evacuation of the water surplus to the outside to avoid any risk of overflow (Figure 1b). The conditions of temperature, pH, and conductivity parameters of the water basin were monitored to ensure that the process of mud colonization by microorganisms, so-called maturation process, took place during the sampling period and ranged between 24 and 30 °C, and 7.2–8.1 and 1010–3115 µS.cm^−1^. Two types of sampling were carried out: (i) from biofilm covering the mud in the maturation basin, and (ii) from epilithic biofilm on the walls of mud basin with the more extensive algae development. 

Samples from the perceptibly mature mud itself and the basin walls collected on 24 July 2014 and 4 August 2014 were globally observed for the first taxonomic investigation based on cell morphology and then cyanobacteria from the mud were inoculated (no later than 3 days after sampling) on solid medium (5 or 10 g·L^−1^ of agar) with medium Z8 and Z8-salt [21]. Isolations were carried out by repeated transfers of single cyanobacteria cells or filaments, on solid or liquid media (at least three times) under an inverted microscope (Nikon ECLIPSE TS100) (Figure 1). Growing clones were then cultured in 25 cm^3^ culture flasks (Nunc, Roskilde, Denmark) containing 10 mL of Z8. Cyanobacteria strains were maintained in the Paris Museum Collection (PMC) [22] at 25 °C, using daylight fluorescent tubes providing an irradiance of 12 μmol photons.m^−2^·s^−1^, with a photoperiod of 16:8 h light:dark. Isolated strains and cultures were all monoclonal and non-axenic. A full taxonomical analysis of the nine cyanobacterial strains was then performed by microscopic and genetic investigation, as described elsewhere [16] and correspond to *Planktothricoides raciborskii* PMC 877.14, *Laspinema* sp. PMC 878.14, *Microcoleus vaginatus* PMC 879.14, *Lyngbya martensiana* PMC 880.14, *Nostoc* sp. PMC 881.14, *Aliinostoc* sp. PMC 882.14, *Leptolyngbya boryana* PMC 883.14, *Dulcicalothrix* sp. PMC 884.14, and *Pseudochroococcus coutei* PMC 885.14. Cultures were transplanted in fresh Z8 liquid medium every six weeks to be maintained in the growing phase when sampled for analyses.

### 2.2. Morphological and Ultrastructural Analyses of Cyanospheres

Morphological analyses of the micro-organisms from the mud and mud basin samples as well as of the Cyanobacteria cultures were carried out using an Axio Imager M2 microscope equipped with an AxioCam MRc Color camera and the ZEN software (ZEISS, Aalen Germany).

The nine cyanobacterial strains and their respective cyanospheres were analyzed by transmission electron microscopy (TEM) as described by Parveen et al. [23], with few modifications [16]. Sample strains were centrifuged, fixed in glutaraldehyde/formaldehyde buffer, washed with Sorensen phosphate buffer, post-fixed with osmium tetraoxide, dehydrated in a graded ethanol series, and then the samples were embedded in EPON resin and sectioned at 0.5 µm, strained with uranyl acetate, and placed on copper plate for observation on TEM (Hitachi HT-7700, Tokyo, Japan). Images were taken using a digital camera (Hamamatsu, Shizuoka Japan). 

Cyanobacterial strains were also analyzed by scanning electron microscopy (SEM). Cultured cyanobacterial cells and/or filaments were centrifuged (10 min; 15,000 rpm), and fixed, dehydrated in a graded ethanol series (50%, 70%, 90%, and 100%) and critical point-dried in liquid CO_2_ (Emitech K850, Quorum Technologies, Laughton, UK), and coated with 20 nm of gold (JEOL Fine Coater JFC-1200) as previously described [16]. The samples were then examined with a Hitachi Scanning Electron SU3500 Premium.

### 2.3. Genomic DNA Extraction

DNA extraction from six biofilm mud samples, from which cyanobacteria were isolated, and from the nine non-axenic cyanobacterial strains, cultured in Z8 medium [24] at 25 °C in 250 mL Erlenmeyer’s vessels, with a photon flux density of 12 µmol.m^−2^·s^−1^ and a 12:12 h light:dark cycle, was carried out with a ZymoBIOMICS DNA mini kit (Zymo Research, Freiburg, Germany) following manufacturer’s protocol. Mechanical lysis was carried out using a bead-beater (TissueLyser II, Qiagen, Hilden, Gemany) for 6 min at maximum speed. An extraction blank was performed as control. DNA quality and quantity was checked with Qubit (Thermo, Whaltam, MA, USA) and nanodrop (SAFAS) apparatus.

### 2.4. Taxonomic Composition of Mud Samples and Cyanosphere

The V4-V5 variable region of the 16S rRNA-encoding gene was amplified from extracted DNA of the nine non-axenic cyanobacterial cultures and six mud samples using 515F and 906R primers [25], and sequenced (Illumina MiSeq paired-end, 2 × 250 bp, GenoScreen, Lille, France). Paired-end reads (representing at least 25,674 raw reads per library) were demultiplexed, quality controlled, trimmed and assembled with FLASH [26]. The obtained sequences, representing at least 14,865 reads per library, were further analyzed using the QIIME 2 2020.11 pipeline [27]. Chimeras were removed and sequences were trimmed to 367 pb and then denoised using the *DADA2* plugin, resulting in Amplicon Sequence Variants (ASVs) [28]. ASVs were affiliated from the SILVA database release 138 using the *feature-classifier* plugin and *classify-sklearn* module [29]. Sequences assigned as Eukaryota, Archaea, mitochondria, chloroplast and unassigned were removed from the dataset and then the sample dataset was rarefied to a list of 1818 sequences. Alpha- and beta-diversity analyses were performed using the MicrobiomeAnalysis platform [30]. Principal coordinates analyses (PCoA) based on unweighted UniFrac distances were performed to examine the dissimilarity of bacterial composition between groups. Among-group variance levels were compared using PERMANOVA (1000 permutations).

### 2.5. Metagenome-Assembled Genomes from Cyanobacteria and Heterotrophs

Total DNA extracts of the nine cyanobacterial strain cultures were sequenced using both 2 × 250 bp Illumina MiSeq 2500 (Nextera XT sample preparation kit) and Single-Molecule Real-Time PacBio RSII platforms (GenoScreen, Lille, France). Scaffolds were assembled from MiSeq and PacBio reads using SPAdes-based Unicycler hybrid-assembler, with default parameters [31,32]. Nodes from assembly graphs were clustered using MyCC (k-mer size = 4, minimal sequence size = 1000) and taxonomically annotated using Contig Annotation Tool [33]. 16S rRNA-encoding genes were extracted from these nodes using Metaxa 2 and annotated using ACT [34]. All contigs were pairwise-aligned using Megablast (E-value ≤ 1.e^−10^), and all sequences sharing a ≥98% similarity on the shortest sequence were considered as coming from the same genome. Congruent data between these diverse methodologies (binning with MyCC and Blast with ACT) allowed characterizing the genomes of each cyanobacterium and its associated heterotrophic bacteria.

The genome assemblies of the cyanobacteria and their respective main co-cultured heterotrophs were integrated in the MicroScope platform v3.14.1 (https://mage.genoscope.cns.fr/microscope/home/index.php, [35]). For each genome, their respective completeness and contamination, as well as their number of CDS and %GC were estimated with CheckM V.1.13 using default parameters [36]. Function annotation was performed using the MicroScope platform using KEGG and MetaCyc databases [37], and results of these annotations were analyzed by PCA, volcano plot with penalized t-test, and heatmap with hierarchical classification (Euclidean distance) using MetaboAnalyst 5 platform [38].

The sequences of Metagenome-Assembled Genomes (MAGs) have been deposited in GenBank under the Bioproject accession number PRJNA686238, PRJNA686242, PRJNA686244, PRJNA686257, PRJNA686258, PRJNA686260, PRJNA686261, PRJNA686262 and PRJNA686263 (Biosample numbers are specified in Appendix A). The nine PMC strains are available from the collection of Cyanobacteria and Microalgae (PMC-ALCP) located in the Muséum national d’Histoire Naturelle (Paris, France, https://mcam.mnhn.fr/fr/collection-de-cyanobacteries-et-microalgues-vivantes-pmc-alcp-470, [22]).

## 3. Results and Discussion

### 3.1. Cyanobacteria Isolates

The thermal mud (a.k.a. the peloid) from Thermes of Balaruc-Les-Bains, one of the oldest thermal centers in France, was mostly colonized by cyanobacteria and microalgae (Figure 1). To characterize the cyanobacteria living in these muds and their potential metabolites production, nine strains representative of the algal community of this environment [14,15] were isolated from the mud biofilms of the retention basin.

Commonly employed cyanobacteria isolation techniques from field samples, such as those presently performed, decrease the number of associated microorganisms, but also preserve microbes strongly attached to the cyanobacterial sheaths [39]. Thus, isolation of cyanobacteria usually results in non-axenic cultures, consisting of reduced microbial consortia, that can be considered as “in vitro blooms” in the sense that the culture conditions provide essential nutrient, temperature, and light supplies for cyanobacteria to survive, grow, or massively proliferate and dominate their microbial environment, as it occurs in nature [40]. Given this, it is relevant to investigate heterotrophic bacteria that are co-isolated with the cyanobacteria in cultures to question their rather opportunistic or mutualistic relationships.

Nine monoclonal strains belonging to the main cyanobacterial taxa colonizing Balaruc’s thermal mud were isolated, taxonomically characterized [16], and then further analyzed for the characterization of their associated microbial communities by metabarcoding and metagenomics. Based on morphological, ultrastructural, and 16S rRNA gene and 16S-23S internal transcribed spacer (ITS) sequence analyses, the nine cyanobacteria were firmly taxonomically identified, the phylogenetic trees obtained from the 16S rRNA gene sequences showing well supported nodes [16]. As previously described, they belong to the orders Oscillatoriales, Nostocales, Chroococcales, Synechococcales and are respectively formerly described as *Planktothricoides raciborskii* PMC 877.14, *Laspinema* sp. PMC 878.14, *Microcoleus vaginatus* PMC 879.14, *Lyngbya martensiana* PMC 880.14, *Nostoc* sp. PMC 881.14, *Aliinostoc* sp. PMC 882.14, *Leptolyngbya boryana* PMC 883.14, *Dulcicalothrix* sp. PMC 884.14, and *Pseudochroococcus coutei* PMC 885.14 (Figure 2).

Interestingly, a similar set of cyanobacteria colonizes thermal mud in other areas, for example, in Abano Thermes in Italy [13], which also include OTU related to similar cyanobacterial taxa including the Microcolaceae, Chroococcales, Oscillatoriaceae, Leptolyngbyaceae orders and the *Phormidium* genus. We notice a global taxonomic similarity (at the level of the most abundant orders) of the different cyanobacterial communities colonizing the thermal muds of Balaruc [16] and Abano [13], together with an apparent temporal stability in Balaruc cyanobacterial mud communities (Figure 1), since several genera were already identified decades ago in muds [14,15]. This suggests that these thermal muds present comparable ecological conditions that select comparable, if not similar, cyanobacteria orders and genera constituting a significant ecological determinant for these established microbial communities.

### 3.2. Cyanosphere Composition

One key metabolic feature of cyanobacteria is autotrophy. Their ability to photosynthesize organic molecules allows excess to be stored, or expelled as carbon-rich exudates consisting mostly of extracellular polysaccharides (EPS) and proteins [41]. In some case, this exudate can build up a sheath around the cyanobacterial cell, offering opportunities for physical anchoring and substantial food supply for heterotrophic bacteria [12].

Microscopic observation of the nine cyanobacteria (Figure 3, Appendix A) confirms the widespread presence of various morphotypes of bacteria, including different *cocci* and *bacilli*/rods, with or without prosthecae, located at the surface of the cyanobacterial cells, depending on the strains. Morphotypes appear firmly attached to external sheath (e.g., *Microcoleus vaginatus* PMC 879.14 or *Nostoc sp.* PMC 881.14), cellular junctions of filaments (e.g., *Leptolyngbya boryana* PMC 883.14) or to the extracellular polysaccharide mucilage (e.g., *Lyngbya martensiana* PMC 880.14, *Planktothricoides raciborskii* PMC 877.14, *Dulcicalothrix* sp. PMC 884.14 or *Pseudochroococcus coutei* PMC 885.14). Because our sample preparation procedure includes different rinsing steps, we assume that these bacteria cells observed on the surface of the cyanobacteria are durably attached or embedded to the extracellular polymeric matrix and may not represent a random association with free-living bacteria that could have contaminated the culture media. This tight physical relationship with cyanobacteria suggests close interactions with cyanobacterial surface components. Besides, the presence of bacteria that are directly attached or immediately adjacent to cyanobacterial cells suggests the possibility of intense and/or specific nutrient exchanges between these microorganisms, as has been observed in other microbial communities dominated by certain cyanobacteria [42].

The observation of heterotrophic bacteria associated with the surface of cyanobacteria had been first interpreted as evidence for symbiotic interactions. For example, the bacteria appeared more numerous and bioactive when positioned on heterocytes of *Anabaena* or *Aphanizomenon* compared with vegetative cells [43]. However, the metabolic interaction between cyanobacteria cells and attached bacteria remains difficult to demonstrate and direct evidence remains rare. Interestingly, the taxonomic comparison of the bacterial communities living attached to *Anabaena* and *Microcystis* cells shows clear differences with the communities occurring in the culture media suggesting that if bacteria living in the immediate vicinity of cyanobacteria may present certain functional specificities potentially related with peculiar interaction it would be regardless to the respective cyanobacteria and heterotroph bacteria genus [44]. Indeed, the different cyanobacteria may differ in many ways regarding their respective sheath and exudate production that might constitute key structures of the cyanobacteria/bacteria interactions, including exopolysaccharides (EPS), that may vary in terms of their amount and composition. EPS are potentially composed of various substances, such as polysaccharides, lipopolysaccharides and glycoprotein heteropolymers and also a various set of metabolites including humic-like substances or secondary metabolites, such as scytonemins, mycosporine-like amino acids (MAAs), or carotenoids with potential photoprotective activities [45]. Although EPS may present some anti-microbial activities [46], it can also specifically serve as primary source of carbon for certain bacteria, such as several Flavobacteria or Roseobacter [47,48].

The bacterial diversity present in Balaruc’s thermal mud and cyanobacterial strain cultures was described by 16S rRNA metabarcoding sequencing (Figure 4). The biofilm mud samples (*n* = 6) yielded between 188 and 129 ASVs, while the cyanobacteria strains (*n* = 9) presented 14 to 37 ASVs, and globally the mud communities exhibited higher diversity (illustrated by Shannon’s index) and different community composition (Figure 4A,B) compared with those detected to cyanobacterial strains. This observation is congruent with the general process of cyanobacterial strains isolation which is expected to reduce the associated diversity with regards to the complex microbial community retrieved in the different mud samples (Figure 4C). It also shows that the different cyanobacteria strains exhibited only rare ASVs in common (Figure 4D). Interestingly, among the large diversity of bacteria present in biofilm mud samples, only few heterotrophic bacteria taxa appeared remarkably shared among different cyanospheres (belonging to proteobacterial orders: Burkholderiales, Rhizobiales, Sphingomonadales, and Caulobacteriales), and appeared not to be the most dominant of the mud biofilms, suggesting host-specific association with each cyanobacterial taxa in culture.

Previous investigation of heterotrophic bacteria associated with non-axenic strains of *Microcystis*, *Dolichospermum*, *Planktothrix*, or *Aphanizomenon* [49], or with *Synechococcus* [8] show the presence of a restricted number of associated taxa comprising Proteobacteria (including Alpha- and Gamma-proteobacteria), Bacteroidetes, Sphingobacteria, Actinobacteria, Cytophagia, or Planctomycetes, with a clear enrichment of certain Bacteroidetes in comparison with the free-living fraction of the cultures. Overall, the relative specificity and low-diversity of bacterial communities associated with cyanobacteria cultures suggest specific selective ecologic or metabolic interactions constrained by both the culture conditions and the vicinity of the cyanosphere itself [48]. However, the different factors determining the development of one bacterium or another within the culture remains to be explored, and could be either an opportunistic or a genuine mutualistic association.

### 3.3. Cyanosphere’s Main Actors Revealed by Metagenomics

Synergistic scaffold binning approach based on both *k*-mer and taxonomic affiliation allows characterizing between two (*Laspinema* sp. PMC 878.14) and five (*Lyngbya martensiana* PMC 880.14) high quality MAGs in each culture sample (size > 2,000,000 pb and completeness > 92%). All sequencing and assembly statistics are summarized in Table 1 and Appendix A. As expected, all cultures presented one single cyanobacterial MAG together with other heterotrophic bacteria MAGs, comprising at least one member of the Proteobacteria, and in some cultures, members of the Bacteroidetes, Gemmatimonadetes, Planctomycetes, Armatimonadetes, or Chloroflexi. Although the MAGs corresponding to Cyanobacteria each fit to a single full genome of a monoclonal genotype, the MAGs of all Proteobacteria, on the other hand, showed inaccurately numerous sequences with very high contamination score (up to 300%) indicating that these MAGs each correspond to different genomes that have been artificially binned within the same MAG because of the global taxonomic affiliation obtained for their respective contigs. These should thus be considered as metagenomes.

This metagenomic approach yields a much lower bacterial diversity compared with metabarcoding in terms of taxa. This apparent discrepancy is consistent with the fact that metagenomics is based on the direct sequencing of abundant DNA, when metabarcoding sequences are obtained after targeted amplification of a 16S rRNA fragment. Thus, metagenomics tends to emphasize the most active and numerous bacteria present within the cultures, while metabarcoding may easily access rarer, and even dead bacteria because of the eDNA resistance and DNA amplification process of the metabarcoding procedure [50]. For this reason, we assume that metagenomic approach provides a more accurate vista of the most abundant community of microorganisms within the cultures and is of course most relevant for functional investigations [51].

Heterotrophic bacteria MAGs identified in the different cyanobacterial strains isolated from Balaruc’s mud cultures belong to taxa already reported to be commonly associated with cyanobacteria in culture or in the environment. Halary and co-workers (2021) have recently shown that the cyanospheres associated to four *Aphanizomenon* strains also exhibit Proteobacteria, Bacteroidetes and Gemmatomonadetes [51]. In addition, metagenomic investigation of various *Microcystis* colonies collected in nine lakes around the world [19] and those of 46 *Microcystis* strains collected from 18 north-American lakes [7] have shown the presence of the same higher-rank bacterial taxa (Proteobacteria/Alphaproteobacteria and Bacteroidetes/Cytophagales) within their respective cyanospheres. Interestingly, Perrez-Carrascal and co-workers [20] have recently shown a remarkably stable bacterial composition among the cyanospheres of 109 single *Microcystis* colonies analyzed by metagenomics, suggesting this as evidence for symbiotic relationships between *Microcystis* and its microbiota, supported by functional complementation among partners based on gene annotation analyses.

### 3.4. Functional Genomics of the Cyanosphere

The functional annotation of the 27 MAGs obtained from the metagenomic dataset was achieved by searching orthologs of protein involved in reference metabolic pathways. To this end, both KEGG and MetaCyc pathway databases were used to better explore the metabolic capabilities of the cyanosphere [19]. The Principal Component Analysis (PCA) based on KEGG and MetaCyc pathway completeness (in %age) displayed the taxon-related fractions of the metagenomes in distinct well-distinguished clusters (Figure 5A,B). These clusters clearly separated cyanobacteria from heterotrophic bacteria, especially Proteobacteria, indicating divergences in their functional gene contents related to the cyanobacterial taxa. The nine points corresponding to the cyanobacteria MAGs were more tightly packed together than those corresponding to Proteobacteria and other heterotrophs, suggesting less variation among pathway contents. Volcano plots generated from molecular pathway completeness show that heterotrophic bacteria, taken together, display much more complete functional pathways than cyanobacteria when using KEGG, while MetaCyc presents the opposite landscape, with much higher completeness scores in the pathways from cyanobacteria metagenomes (Figure 5C,D). These observations were further confirmed on heatmap visualization, as KEGG indicates that significantly much more complete pathways were observed in heterotroph, while MetaCyc differently presents more complete pathways in cyanobacteria (Figure 5E,F).

The KEGG and MetaCyc projects (237 and 296 super pathways, respectively) have developed large metabolic pathway databases that are used for a variety of applications including genome analysis and metabolic engineering [37]. On one side, MetaCyc contains many pathways not found in KEGG, from plants (such as many photosynthesis-related processes), fungi, metazoan, and actinobacteria, while, on the other side, KEGG contains pathways not found in MetaCyc, for xenobiotic degradation, glycan metabolism, and metabolism of terpenoids and polyketides [37]. Surprisingly, the functional exploration of the cyanobacterial genomes and their respective cyanosphere through metagenome functional pathway annotation illustrates this obvious discrepancy. Indeed, it suggests that KEGG would be more appropriate for illustrating the metabolism of heterotrophs, while MetaCyc seems to present a better coverage of the specific metabolism of autotrophs, such as cyanobacteria.

Cook and co-workers [19] were recently able to investigate the functional metagenomes of bacteria associated with *Microcystis* colonies using KEGG interrogation. These authors pointed out that the KEGG database does not contains many *Microcystis* functions relevant to photosynthesis processes. They also established that the numerous key pathways involved in anaerobiosis were present in heterotroph metagenomes, suggesting that anaerobic process may play an important role in the nutrient recycling within the cyanosphere. Interestingly, our results also support this assumption highlighting such potential metabolic complementation on potential nutrient recycling within the cyanosphere through anaerobic metabolism capability of heterotrophs.

Perez-Carrascal and co-workers [20] also investigated gene functions in both *Microcystis* and its cyanosphere using orthologous search based on KEGG interrogation. Since they exhibit more similar functional gene contents, phylogenetically related MAGs tend to cluster together. We presently observed a similar pattern for cyanobacteria and proteobacteria clusters that discriminate from other heterotroph bacteria. As expected for distantly related bacteria, cyanobacteria and associated heterotrophs exhibit different functional gene repertoires, some of which could be complementary and mutually beneficial. This observation also suggests that the different members of these clusters (i.e., cyanobacteria, proteobacteria, and other bacteria) may present a certain functional redundancy because of their respective global similarity in terms of functional gene contents. 

Taken together, these functional annotation data (Appendix A) showed a clear distinction between metabolic pathways found in cyanobacteria and in their heterotrophic partners, and a global homogeneity with little variations in the metabolic potential within these respective groups, which seems mostly related to specificities of autotrophic versus heterotrophic metabolisms. Specifically, certain functional pathways display higher completion levels in heterotrophs (considered together) compared with cyanobacteria including the amino acid, carbohydrate (except the gluconeogenesis) and energy metabolisms (except oxidative phosphorylation). On the contrary, cyanobacteria appear more enriched in protein functions related to energy production (except the lipid degradation), nitrogen fixation, and biosynthesis of amino acids, carbohydrates, electron carriers, secondary metabolites and co-factors, such as various carotenoids, vitamin E, and vitamin K2.

Recently, Pascault and co-workers [52] explored the respective metabolic roles of the cyanobacteria and their bacterial microbiota in a day-night environmental meta-transcriptomic analysis of the cyanosphere supported by KEGG annotations. This study revealed that the functional expression of the active community was overall driven by their respective trophic modes. Indeed, the autotrophs (*Dolichospermum* and *Microcystis*) exhibited functional enrichments in photosynthesis, CO_2_ fixation, carbohydrate metabolism, oxidative phosphorylation, N_2_ fixation, phosphorus and glutamate metabolisms, while heterotrophs mostly presented enrichments in carbohydrate metabolism, TCA cycle, glycolysis/gluconeogenesis, pyruvate metabolism, and transcription. Taken together, this approach exquisitely illustrated the global metabolic complementation that exists between autotrophs and heterotrophs within the cyanosphere and the global functional complementarity between the different taxa.

Photosynthesis-derived carbon produced by cyanobacteria is exported and forms a thick extracellular polymeric substance (EPS) that likely fuels the growth of heterotrophic bacteria that make use of these compounds [53]. Indeed, numerous pathways of carbohydrate degradation have been retrieved in the metagenomes of the bacteria living within *Microcystis* colonies maintained with extracellular polysaccharidic mucilage [19]. Nitrogen is also considered as a limiting nutrient in aquatic ecosystems and the capability of fixing atmospheric N_2_ by certain cyanobacteria may support higher production of the cyanosphere communities under limiting conditions [52]. Carotenoids may also constitute a remarkable example of common good for the cyanosphere community as they could be beneficial for the cyanobacteria by broadening the photosynthetic light absorption spectrum, acting as an accessory pigment, while also contributing to the photoprotection of heterotrophic bacteria against oxidative stress [20,54].

Taken together, these observations suggest that the potential metabolic association of cyanobacteria and heterotrophic bacteria within the cyanosphere might be driven by the reciprocal exchange of common goods, including carbon and nitrogen sources, as well as various vitamins and co-factors.

## 4. Conclusions

The cyanospheres of Balaruc’s mud cyanobacterial cultures were composed of distinct bacterial ASVs presenting similar functional potential highlighting functional redundancy, which could constitute an advantage for the respective culture fitness through metabolic exchanges and recycling. Besides these findings, our results also highlighted the great influence of the reference protein database chosen for performing functional annotation of the metagenomes from organisms of the cyano-/phyco-sphere and the difficulty of selecting one unique database that appropriately cover both autotroph and heterotroph metabolic specificities.

Future efforts concerning the functional annotation of the cyanosphere should investigate multiple databases to cover as many genes and pathways as possible, as no unique current database seems to contain all metabolic pathways potentially present in complex microbial assemblages, such as those suspected to occur within the cyanosphere.

## Figures and Tables

**Figure 1 microorganisms-10-02337-f001:**
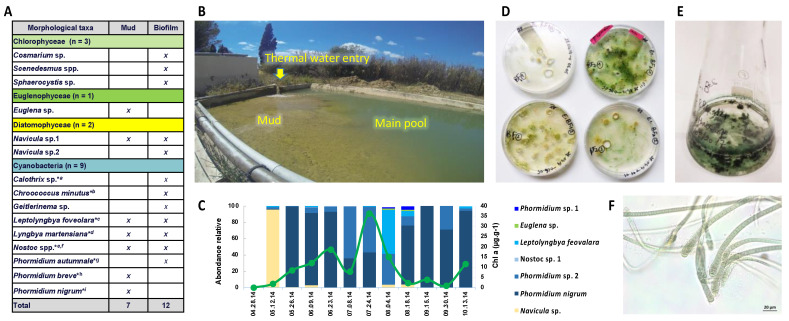
The algae and cyanobacteria community colonizing the Balaruc’s thermal mud. (**A**) The micro-algae and cyanobacteria present in surface biofilm and in the mud of the maturation basin of Balaruc’s thermal mud. * indicates that this initial taxonomic identification was further re-evaluated based on polyphasic analyses of the nine strains, accordingly: a = *Dulcicalothrix* sp., b = *Pseudochroococcus coutei*, c = *Leptolyngbya boryana*, d = *Lyngbya martensiana*, e = *Nostoc* sp., f = *Aliinostoc* sp., g = *Microcoleus vaginatus*, h = *Laspinema* sp., i = *Planktothricoides raciborskii*. Remark: Monoclonal strains were obtained for all cyanobacteria taxa except for *Geitlernema*, which we did not succeed to maintain in culture. (**B**) General view of the mud maturation basin during the sampling (24 July 2014). (**C**) Monitoring of the mud algal and cyanobacterial succession during 2014 spring and summer seasons. (**D**) Solid medium culture of various cyanobacteria (presently *Calothrix*) collected from the Balaruc’s thermal mud (24 July 2014). (**E**) Liquid media culture of a monoclonal strain. (**F**) Microscopic view of the *Dulcicalothrix* strain PMC 884.14.

**Figure 2 microorganisms-10-02337-f002:**
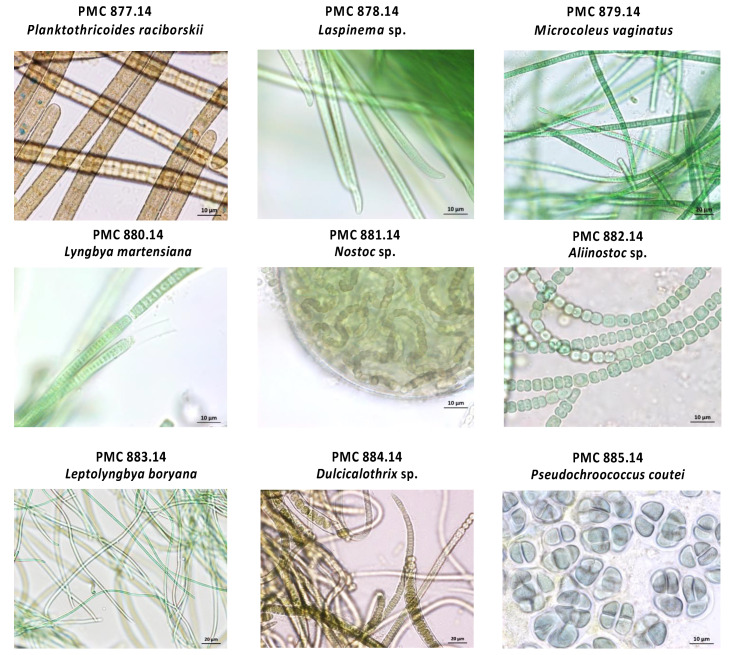
Representative microscopic view of the nine strains from the thermal mud of the Thermes of Balaruc-Les-Bains.

**Figure 3 microorganisms-10-02337-f003:**
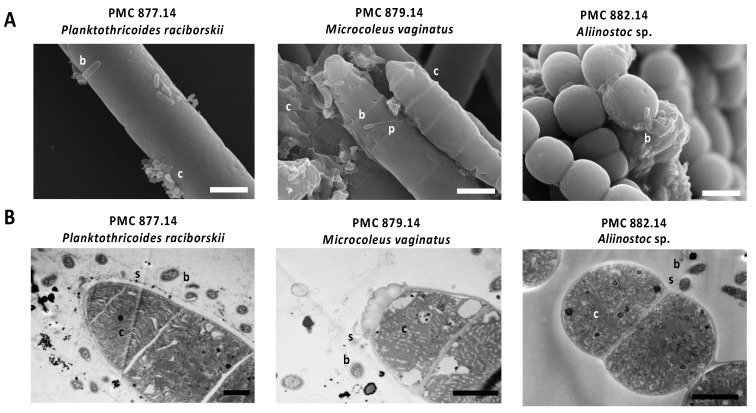
Representative scanning (**A**) and transmission (**B**) electron micrographs of several cyanobacterial strains isolated from Balaruc’s thermal mud showing the presence of various bacteria morphotypes. (**A**) The different *cocci* and *bacilli* and prosthecea indicated with c, b, and p, respectively, been attached on the surface of the cyanobacterial cells. (**B**) Cyanobacteria cells, bacteria and sheath are indicated by c, b, and s, respectively. Scale bars represent 2 and 1 µm for A and B, respectively. Representatives SEM and TEM pictures of the nine strains are also observed on Appendix A.

**Figure 4 microorganisms-10-02337-f004:**
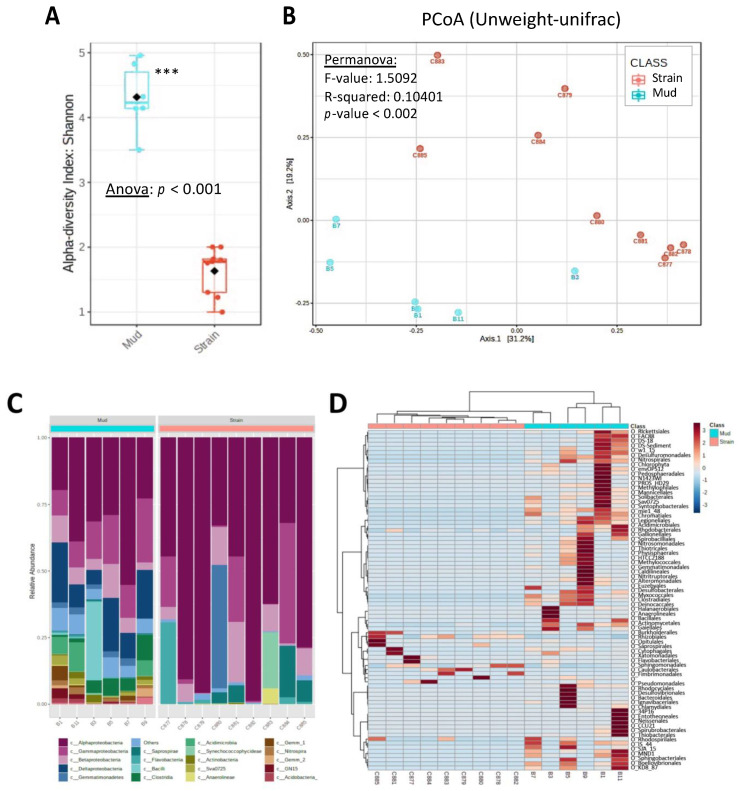
Bacterial diversity of the Balaruc’s biofilm muds (*n* = 6, turquoise) and cyanobacterial strains (*n* = 9, pink) by 16S rRNA metabarcoding. (**A**) Alpha diversity analyzed with Shannon’s index. (**B**) Beta diversity analyzed on PCoA performed with unweight-Unifrac distance and Permanova analysis. (**C**) Relative proportion of all principal heterotroph bacterial orders. C877–C885 indicates PMC 877–885 strains. (**D**) Heatmap with hierarchical classification (Euclidean distance) of dominant ASV (<0.1%), with the color scale representing respective abundancies. *** indicate Anova *p*-value < 0.001.

**Figure 5 microorganisms-10-02337-f005:**
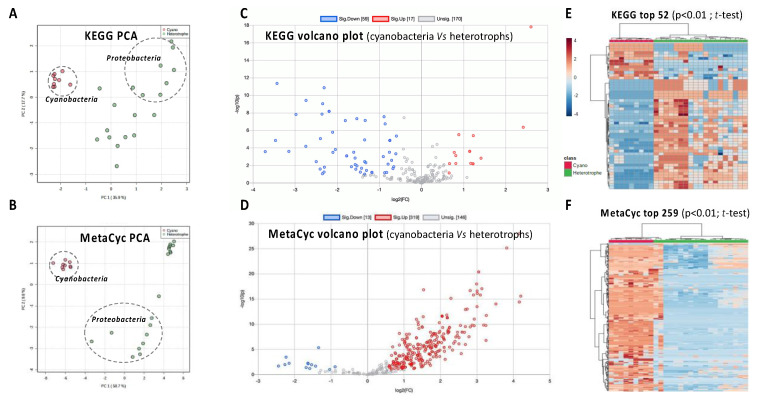
Functional metagenomics of the cyanosphere from the nine cyanobacterial strain and their cyanosphere according to KEGG (**A**,**C**,**E**) and MetaCyc (**B**,**D**,**F**) database interrogation. (**A**) Principal Component Analysis based on KEGG pathway completeness (%age) in cyanobacteria and heterotroph metagenomic sequences. (**B**) Principal Component Analysis based on MetaCyc pathway completeness (%age) in cyanobacteria and heterotroph metagenomic sequences. Dotted ellipse indicates Cyanobacteria and Proteobacteria metagenomes from the nine cyanobacterial strains. (**C**) Volcano plot of individual KEGG pathways completeness (%age) comparing cyanobacteria vs. heterotroph bacteria metagenomes. (**D**) Volcano plot of individual MetaCyc pathways completeness (%age) comparing cyanobacteria vs. heterotroph bacteria metagenomes. Functional pathway significantly over- and under-represented in cyanobacteria are indicated in red and blue, respectively. (**E**) Heatmap with hierarchical classification of KEGG pathway completeness (relative %age indicated by color scale). (**F**) Heatmap with hierarchical classification of MetaCyc pathway completeness (%age). Cyanobacteria and heterotroph metagenomes are indicated in red and green, respectively.

**Table 1 microorganisms-10-02337-t001:** Cyanosphere associated-MAG statistics and taxonomic annotations (cyanobacteria in bold, additional information comprising checkM completeness and contamination and taxonomical affiliation are indicated on Appendix A). MAG with the same PMC number were obtained from the same strains accordingly.

Strain__MAG (16S Phylum Annotation)	nb_contigs	Size (pb)	GC%	nb CDSs	Microscope Affiliation
PMC 877__Bacteroidetes	8	2,940,501	32.92	2707	Bacteroidetes
**PMC 877__Cyanobacteria**	**15**	**7,391,557**	**43.50**	**7130**	** *Planktothricoides* **
PMC 877__Proteobacteria	7	3,625,685	64.63	3616	Sphingomonadales
**PMC 878__Cyanobacteria**	**69**	**7,336,742**	**47.37**	**5829**	** *Laspinema* **
PMC 878__Proteobacteria	45	12,013,276	65.81	11,940	Proteobacteria
**PMC 879__Cyanobacteria**	**34**	**6,900,942**	**45.68**	**6400**	** *Microcoleus* **
PMC 879__Proteobacteria	11	10,840,780	67.92	10,627	Proteobacteria
PMC 880__Bacteroidetes	1803	14,271,057	48.22	14,012	Bacteroidetes
**PMC 880__Cyanobacteria**	**132**	**6,444,104**	**39.60**	**5904**	** *Lyngbya* **
PMC 880__Gemmatimonadetes	25	4,542,236	65.84	4112	Gemmatimonadetes
PMC 880__Planctomycetes	216	6,159,044	53.18	5068	Planctomycetes
PMC 880__Proteobacteria	2151	21,578,483	64.87	22,250	Proteobacteria
**PMC 881__Cyanobacteria**	**648**	**7,938,353**	**41.66**	**7711**	** *Nostoc* **
PMC 881__Proteobacteria	1146	27,227,936	67.65	25,649	Proteobacteria
PMC 882__Armatimonadetes	5	7,360,037	54.92	6711	Armatimonadetes
**PMC 882__Cyanobacteria**	**46**	**8,132,153**	**41.34**	**7582**	** *Trichormus* **
PMC 882__Proteobacteria	41	9,857,661	66.03	9926	Proteobacteria
PMC 883__Armatimonadetes	4	3,575,512	55.83	3399	Fimbriimonadales
PMC 883__Chloroflexi	6	5,641,484	58.03	5032	Chloroflexi
**PMC 883__Cyanobacteria**	**5**	**6,694,777**	**46.94**	**6313**	** *Leptolyngbya* **
PMC 883__Proteobacteria	72	6,645,423	67.63	6660	Proteobacteria
PMC 884__Bacteroidetes	4	2,942,276	39.52	3054	Bacteroidetes
**PMC 884__Cyanobacteria**	**46**	**13,243,152**	**38.60**	**11,176**	** *Calothrix* **
PMC 884__Proteobacteria	132	11,961,935	68.16	12,136	Proteobacteria
PMC 885__Bacteroidetes	30	7,094,005	39.52	6347	*Flavobacterium*
**PMC 885__Cyanobacteria**	**137**	**5,858,006**	**35.29**	**5328**	** *Chroococcus* **
PMC 885__Proteobacteria	52	18,208,582	65.84	17,483	Proteobacteria

## Data Availability

The MAG sequences have been deposited in GenBank under the Bioproject accession number PRJNA686238, PRJNA686242, PRJNA686244, PRJNA686257, PRJNA686258, PRJNA686260, PRJNA686261, PRJNA686262 and PRJNA686263.

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
