# Peer review of "Metagenome-Based Exploration of Bacterial Communities Associated with Cyanobacteria Strains Isolated from Thermal Muds"

_microorganisms, 2022, doi:10.3390/microorganisms10122337_

Round 1

Reviewer 1 Report

The paper is well written, organized and adds new understanding to the study of the cyanosphere. The methods used are quite modern and their combination is relevant. But there are some minor comments on the manuscript.

-          The work studies the microorganisms of the thermal mud of Balaruc-les-Bains (France), but there is no information on the chemical parameters of this ecosystem, as well as temperature;

-          What is the rationale for choosing the 16S rRNA gene region for taxonomic identification of microorganisms using Illumina technology? Why not V3-V4 or V4, for example?

-          Figures 2 (A) and 5 (C,D) are of low quality and difficult to read. Table 1 has a lot of information. Is it possible to reduce it by choosing the most important. Otherwise, you can transfer it to supplementary material.

The very good SEM and TEM images of cyanobacteria should be noted.

After correcting small but important remarks and improving the quality of images, the article is recommended for publication in the journal.

Author Response

Comments and Suggestions for Authors

The paper is well written, organized and adds new understanding to the study of the cyanosphere. The methods used are quite modern and their combination is relevant. But there are some minor comments on the manuscript.

Answer: We thank the reviewer for his(her) positive general comments on our MS and have addressed point-by-point answers to his(her) specific remarks.

-          The work studies the microorganisms of the thermal mud of Balaruc-les-Bains (France), but there is no information on the chemical parameters of this ecosystem, as well as temperature;

Answer: We thank the reviewer for this remark and have now added physico-chemical parameter of the sampling locations in the M&M section.

“The retention basin (500 m3) maintains a water level above the mud and the continuous evacuation of the water surplus to the outside to avoid any risk of overflow (Figure 1d). The conditions of temperature, pH and conductivity parameters of the water basin were monitored to ensure that the process of mud maturation takes place during the sampling period and rang between 24-30°C, 7.2-8.1 and 1010-3115 µS.cm-1.”

-          What is the rationale for choosing the 16S rRNA gene region for taxonomic identification of microorganisms using Illumina technology? Why not V3-V4 or V4, for example?

Answer: Indeed, the reviewer is right when questioning the representativity of the value of the V4-V5 region sequenced in the metabarcoding analysis. Actually, it is already an old debate and no 16S region appears absolutely resolutive and the selection of the amplified region still constitutes a compromise of taxonomic universality and selectivity. For this purpose, the V4-V5 region is routinely used for global for bacterial metabarcoding investigations and we assume to use it for the rational evaluation the bacterial diversity of Balaruc’s muds and strain cultures.

-          Figures 2 (A) and 5 (C,D) are of low quality and difficult to read. Table 1 has a lot of information. Is it possible to reduce it by choosing the most important. Otherwise, you can transfer it to supplementary material.

Answer: Thank you for these remarks, the fig 2 and 5 have been modified accordingly. Table 1 has been also reduced and supporting information transferred to supplementary material.

The very good SEM and TEM images of cyanobacteria should be noted.

Answer: Thank you for this remark. In order to highlight most illustrative picture, best images have been selected, and others transferred to supplementary materials.

After correcting small but important remarks and improving the quality of images, the article is recommended for publication in the journal.

Answer: Improvement have been now provided accordingly, and hope that our MS is now suitable for publication.

Reviewer 2 Report

The study explores cyanobacterial strains isolated from a thermal mud and their associated microbiota or cyanosphere. DNA sequencing allowed the characterization of the microbiota that remained associated with cyanobacteria after isolation and the comparison with the composition of the microbial community from the original environment. This is an interesting approach to understand the physiology of cyanobacteria in non-axenic culture and the possible selection of components of its cyanosphere. The study may improve if the authors further explore the results of the cyanosphere characterization and the microbial communities in the environment, such as host(cyanobacteria)-specificity, complementarity functions, production of common goods, mutualistic relationships, which are cited in the abstract and conclusion but were not clearly shown or deepen in the results section. On the contrary, the analyses are very generic in terms of metabolic functions and the discussion follows this path.

abstract

Thus, the physical proximity, host-specificity and complimentary functions advocate for their complementarity

complementary correct and avoid repetition

Introduction 

78

we aimed to characterized

correct

2. Materials and methods

Please include in this section the period of time from sampling and strain isolation (2014) to the time when the analyses were carried out (microscopy, metagenomics etc) 

88

Mud and biofilm samples were collected in 2014

in the subsequent sections the authors refer to isolated cyanobacterial strains and mud samples

Were the strains isolated from mud or biofilm samples? (98-103)

All over the manuscript the term mud samples is used but there is no explanation on the difference between biofilm and mud samples, or biofilm covering the mud and epilithic biofilm, when these samples were used, if they were compared and why etc

2.3. Genomic DNA extraction

no biofilm samples? no other analysis cited the biofilm samples

104

12 µmol photons.cm-2.s-1,

m2

124

Each cyanobacterial strains

Cyanobacterial strains

2.6. Metagenome sequencing and assembly of cyanobacteria’s strains microbial consortia

From the description it seems that this refers to

Metagenome sequencing and assembly of cyanobacteria strains AND THEIR microbial consortia

Total DNA extracts of the nine cyanobacterial strain cultures were sequenced using

How were the libraries prepared?

153

Paired-end reads (representing at least 14,865 raw reads per libraries...obtained sequences, representing at least 25,674 reads per library

How can you obtain 25,674 reads per library after the treatment of 14,865 raw reads per library?

Results and discussion

200

nine strains representative of the algal community of this environment were isolated from the water column and biofilms of the retention basin collected 201 the 07-24-2014 and the 08-04-2014

This information should appear in Methods

Methods did not refer to water column samples, in that section the authors cite "mud samples", but they were not properly described 

209

growth

grow

3.1. Cyanobacteria isolates and their associated bacterial heterotrophs

Differently from the subtitle, this section only describes cyanobacterial strains, not the associated bacteria

224 and Fig 2

Nine monoclonal strains belonging to the different cyanobacterial taxa that colonize  Balaruc’s thermal mud were selected for taxonomic characterization

Aren't these the same strains described in Duval et al. FEMS Microbes, Volume 2, 2021?

If so, there is no need to describe their isolation or show the classification based on 16SRNA again. Section 3.1 is questionable.

237

We notice a global taxonomic similarity of the different cyanobacterial communities colonizing the thermal muds of Balaruc [16] and Abano [13]

In terms of orders? Can the authors state that from the characterization of only 9 strains? Same observation for the conclusion at the end of this paragraph.

Fig 4 (C, cyanobacteria; C, sheath; S sheath

282

bacteria living in the immediate vicinity of cyanobacteria may present certain specificities potentially related with peculiar interaction with the cyanobacteria regardless to the cyanobacteria genus [46].

Ref 46 cites the opposite, that bacterial community differs according to the cyanobacterial genus, as well as ref 51

303

The thermal muds (n = 6) yield between 188 and 129 ASVs, when the strains (n = 9) present 14 to 37 ASVs

Please clarify if the “thermal muds” represent samples collected at the same time as the cyanobacterial strainsthermal muds are mud samples (?) and the doubt on biofilm, water column etc perists

The method section states that 2 types of sampling were carried out: i) from biofilm covering the mud and from epilithic biofilm on the walls of mud basin

Is is not clear which mud samples were characterized by 16SrRNA analysis

same for fig 5

Fig 5 low resolution

311

Several heterotrophic bacteria taxa are shared among different cyanospheres …or shared with mud samples, suggesting host-specific association with each cyanobacterial taxa 

Contradictory statement, please change. Shared or different taxa in the cyanospheres were not evidenced in the figure. 

315 - 326

This discussion is vague, which point do the authors want to show? Specific associations of heterotrophic bacteria with certain cyanobacteria? This is a list of several taxa associated with other cyanobacteria. What is being compared? Why cite a diatom?

I suggest that the authors focus their results in those taxa shared between mud samples and cyanobacteria. If they were collected on the same date, did some taxa remain in culture since isolation, or else, the associated bacterial community changed entirely. Are there more lasting taxa? The 9 cyanobacterial strains represent different genera, present different morphologies and physiologies, are these differences reflected in their cyanospheres? In sum, it is not clear why the comparisons shown in fig 5 were carried out, which conclusions can be drawn, please discuss and modify.

Table 1. Do the samples with the same numbers  (877, 878) refer to genomes recovered from the same culture? indicate

355-364

most active - please clarify, abundant, metabolic active?

general: change the term when for while

472-483

This discussion cites previous knowledge but does not relate to the findings of this study, since here, specific pathways were not explored in the results.  

The conclusion in 484 487 is vague, the authors did not use their results to state this, this is previous knowledge.

The same applies to the conclusion, which summarizes previous known facts and not specific results from this study (1st and last paragraphs). For example:

Their metagenomes encode some functions that are complimentary and may promote the growth of cyanobacteria, 

no such function was highlighted in the results

the physical proximity, host-specificity and complementary functions advocate for a mutualist relationship between cyanobacteria hosts and their associated microbiota.

no specific function related to mutualism was highlighted in the results

complimentary

complementary 

Author Response

Reviewer2

Comments and Suggestions for Authors

The study explores cyanobacterial strains isolated from a thermal mud and their associated microbiota or cyanosphere. DNA sequencing allowed the characterization of the microbiota that remained associated with cyanobacteria after isolation and the comparison with the composition of the microbial community from the original environment. This is an interesting approach to understand the physiology of cyanobacteria in non-axenic culture and the possible selection of components of its cyanosphere. The study may improve if the authors further explore the results of the cyanosphere characterization and the microbial communities in the environment, such as host(cyanobacteria)-specificity, complementarity functions, production of common goods, mutualistic relationships, which are cited in the abstract and conclusion but were not clearly shown or deepen in the results section. On the contrary, the analyses are very generic in terms of metabolic functions and the discussion follows this path.

Answer: We thanks the reviewer for his(her) constructive remarks. We have now provided a point-by-point to the following comments and modified our MS accordingly.

Abstract

Thus, the physical proximity, host-specificity and complimentary functions advocate for their complementarity => complementary correct and avoid repetition

Answer: This has been modified accordingly.

Introduction 

78: we aimed to characterized => correct

Answer: This has been modified.

  1. Materials and methods

Please include in this section the period of time from sampling and strain isolation (2014) to the time when the analyses were carried out (microscopy, metagenomics etc).

Answer: This information has been added in the M&M section accordingly.

88 : Mud and biofilm samples were collected in 2014

in the subsequent sections the authors refer to isolated cyanobacterial strains and mud samples

Were the strains isolated from mud or biofilm samples?

Answer: The different strains were isolated from the mud samples. This aspect has now been clarified, thank you.

98-103: All over the manuscript the term mud samples is used but there is no explanation on the difference between biofilm and mud samples, or biofilm covering the mud and epilithic biofilm, when these samples were used, if they were compared and why etc

Answer: Actually, the present is mostly focused on mud community but we have also investigated the microbial communities that were also present on the walls of the basin as it may constitute a connected compartment in which micro-organisms may transit too. This has now been clarified in order to avoid misleading the rider on this aspect.

2.3. Genomic DNA extraction

no biofilm samples? no other analysis cited the biofilm samples

Answer: Thank you for this remark. This information has been added in the M&M

104: 12 µmol photons.cm-2.s-1 => m2

Answer: Thank you for this remark. This has been corrected.

2.6. Metagenome sequencing and assembly of cyanobacteria’s strains microbial consortia

From the description it seems that this refers to

“Metagenome sequencing and assembly of cyanobacteria strains and their microbial consortia”…..

“Total DNA extracts of the nine cyanobacterial strain cultures were sequenced using”

=> How were the libraries prepared?

Answer: Libraries were prepared by the Genoscreen company with Nextera XT sample preparation kit following manufactor’s recommendations. This information has been added to the M&M.

153: Paired-end reads (representing at least 14,865 raw reads per libraries...obtained sequences, representing at least 25,674 reads per library

How can you obtain 25,674 reads per library after the treatment of 14,865 raw reads per library?

Answer: Thank you for this remark. The numbers have been here inverted and this has now been corrected in the text.

Results and discussion

200

nine strains representative of the algal community of this environment were isolated from the water column and biofilms of the retention basin collected the 07-24-2014 and the 08-04-2014

This information should appear in Methods

Answer: This information has now been transferred to M&M accordingly.

Methods did not refer to water column samples, in that section the authors cite "mud samples", but they were not properly described 

Answer:  We are sorry if this information would have been somehow confusing. This point has now been clarified.

209: growth => grow

Answer: This has been corrected accordingly

3.1. Cyanobacteria isolates and their associated bacterial heterotrophs

Differently from the subtitle, this section only describes cyanobacterial strains, not the associated bacteria

Answer: Thank you for this remark. This subtitle has now been corrected accordingly.

224 and Fig 2

Nine monoclonal strains belonging to the different cyanobacterial taxa that colonize Balaruc’s thermal mud were selected for taxonomic characterization

Aren't these the same strains described in Duval et al. FEMS Microbes, Volume 2, 2021?

If so, there is no need to describe their isolation or show the classification based on 16SRNA again. Section 3.1 is questionable.

Answer: Indeed, those are the same nine strains than in Duval et al. 2021 in which in-depth taxonomic description of the strains have been performed. To avoid redundancy with this work, we decided to remove the taxonomical description of the strains and more simply refer to this work.

237: We notice a global taxonomic similarity of the different cyanobacterial communities colonizing the thermal muds of Balaruc [16] and Abano [13]

In terms of orders? Can the authors state that from the characterization of only 9 strains? Same observation for the conclusion at the end of this paragraph.

Answer: Indeed, this assumption is based on the observation of most abundant taxa within Balaruc’s and Abano’s thermal muds, which present many taxonomic composition similarities, that corresponds to the cyanobacteria orders on which the efforts of isolation have been especially focused on. This point has now been emphasized in the text.

bacteria living in the immediate vicinity of cyanobacteria may present certain specificities potentially related with peculiar interaction with the cyanobacteria regardless to the cyanobacteria genus [46].

Ref 46 cites the opposite, that bacterial community differs according to the cyanobacterial genus, as well as ref 51

Answer: Thank you for this remark. We fully share the idea that bacterial community associated with the different cyanobacteria strains present specific bacterial communities as previously shown by Louati et al. 2015 and Zhu et al. 2016. We have now reformulated these sentences in order to avoid any misleading of the reader concerning this specific point, accordingly.

303: The thermal muds (n = 6) yield between 188 and 129 ASVs, when the strains (n = 9) present 14 to 37 ASVs

Please clarify if the “thermal muds” represent samples collected at the same time as the cyanobacterial strainsthermal muds are mud samples (?) and the doubt on biofilm, water column etc perists

Answer: Thank you for this remark. This terminology has now been unified along all the text in order to avoid confusing the reader.

The method section states that 2 types of sampling were carried out: i) from biofilm covering the mud and from epilithic biofilm on the walls of mud basin

Is is not clear which mud samples were characterized by 16SrRNA analysis

Answer: Actually, only biofilm mud samples have been investigated for microbial diversity (and cyanobacteria isolation). This information has now been clarified and unified in the M&M and result sections.

same for fig 5

Fig 5 low resolution

Answer: This figure has been now improved accordingly.

311: Several heterotrophic bacteria taxa are shared among different cyanospheres …or shared with mud samples, suggesting host-specific association with each cyanobacterial taxa 

Contradictory statement, please change. Shared or different taxa in the cyanospheres were not evidenced in the figure. 

Answer: Thank you for this remark. This important point has now been clarified and emphasized.

315 – 326: This discussion is vague, which point do the authors want to show? Specific associations of heterotrophic bacteria with certain cyanobacteria? This is a list of several taxa associated with other cyanobacteria. What is being compared? Why cite a diatom?

I suggest that the authors focus their results in those taxa shared between mud samples and cyanobacteria. If they were collected on the same date, did some taxa remain in culture since isolation, or else, the associated bacterial community changed entirely. Are there more lasting taxa? The 9 cyanobacterial strains represent different genera, present different morphologies and physiologies, are these differences reflected in their cyanospheres? In sum, it is not clear why the comparisons shown in fig 5 were carried out, which conclusions can be drawn, please discuss and modify.

Answer: Thank you for these remarks. This discussion section has now been shortened and reworded in order to better highlight the main ideas. Diatom’s citation has been removed.

Table 1. Do the samples with the same numbers (877, 878) refer to genomes recovered from the same culture? Indicate

Answer: Indeed, this has been specified in the table legend.

355-364: most active - please clarify, abundant, metabolic active?

Answer: “most active “ has been replaced by “most abundant”.

general: change the term when for while

Answer: This has been corrected accordingly.

472-483: This discussion cites previous knowledge but does not relate to the findings of this study, since here, specific pathways were not explored in the results.  

The conclusion in 484 487 is vague, the authors did not use their results to state this, this is previous knowledge.

Answer: We have now improved this discussion section in order to better articulate references with present results.

The same applies to the conclusion, which summarizes previous known facts and not specific results from this study (1st and last paragraphs). For example:

Their metagenomes encode some functions that are complimentary and may promote the growth of cyanobacteria, 

no such function was highlighted in the results

Answer: Thank you for this remark. Conclusion has been modified accordingly.

the physical proximity, host-specificity and complementary functions advocate for a mutualist relationship between cyanobacteria hosts and their associated microbiota.

no specific function related to mutualism was highlighted in the results

Answer: Conclusion has been modified accordingly.

Complimentary => complementary 

Answer: This has been corrected.

Round 2

Reviewer 2 Report

The manuscript has been modified according to the comments and has improved. Some details can be corrected as listed below.

116 evacuation of the water surplus to the outside to avoid any risk of overflow (Figure d)

Fig 1 B

119 maturation takes place

took place

119 during the sampling period. Temperature rang between 24-30°C, pH, 7.2-8.1 and conductivity 1010-3115 µS.cm-1 120 .

please explain why these variations indicate “maturation”

were inoculated on solid medium with medium Z8 and Z8-salt [21]. Isolations were carried out 

maybe cite fig 1

123 Samples from the mud itself and the basin walls collected the 07-24-2014 and the 08- 124 04-2014

Do these dates correspond to “mud maturation”? why were they chosen?

How long were these samples maintained and processed before the described analysis?

Fig 1

  1. please cite that the cyanobacteria shown in the table were explored in this study

E) Liquid media culture of a monoclonal strain. 

of which genus?

F) Microscopic view of the Dulcicalothrix strain PMC 884.14

not necessary

155 Each cyanobacterial strains were

cyanobacterial strains were

169 DNA quality and quantity was checked with Qubit (Thermo)

only quantity, quality with spectrophotometer, Nanodrop or similar

164 DNA extraction from six biofilm mud samples from which cyanobacteria were 165 isolated and from the nine non-axenic cyanobacterial strains

X

189 gene was amplified from extracted DNA of the nine non-axenic cyanobacterial 190 cultures and five mud samples

6 or 5 mud samples? there are 6 in fig 5

163 2.3. Genomic DNA extraction

described again in 187

172 2.4. Molecular phylogeny of cyanobacterial strains

no need to describe this, already published in Duval et al [16]

in the present study rRNA data was obtained as cited - 215 annotated using Contig Annotation Tool [35]. 16S rRNA-encoding genes were 216 extracted

184 2.5. Taxonomic composition of mud samples and cyanosphere

V4-V5 variable region of the 16S rRNA-encoding 189 gene was amplified from extracted DNA of the nine non-axenic cyanobacterial 190 cultures and five mud samples 

2.6. Metagenome sequencing and assembly of cyanobacteria’s strains microbial consortia

these subtitles can be confusing

please emphasize in 2.6 subtitle that this part refers to Metagenome-Assembled Genomes from cyanobacteria and heterotrophs

while in 2.5 it refers to 16S amplicon sequence to characterize the community composition or 16S rRNA metabarcoding sequencing

252 also preserved

preserve

250-260 

appropriate to cite the time elapsed between collection and isolation of sample/strains and analysis 

346-347

unclear statement

373 and 379

and cyanobacterial strain cultures

figure 4 C) only heterotrophs?

D) ASV heatmap (<0.1%) 

?

383-389 interestingly repeated

437 and 440 while metabarcoding 

Table 1 suggestion maintain a more detailed taxonomic affiliation, not just Phylum 

479. Differently

delete

483-487

unclear statement

495 exploration of the cyanosphere

and cyanobacterial genomes

510 Our results also support this assumption highlighting such potential metabolic 511 complementation within the cyanosphere

which assumption, that KEGG does not contain functions relevant to photosynthesis processes?

and why this  highlights potential metabolic complementation within the cyanosphere ?

514 According to their respectively more similar functional gene contents,?

Since they have similar gene contents (?)

520 suggests that the different members of these clusters

within each cluster

Fig 5  

Functional metagenomics of the cyanosphere from the nine cyanobacterial strains

Functional metagenomics of the cyanobacterial strains and their cyanosphere  

C) and D) of comparing cyanobacteria and heterotrophic bacteria metagenomes taken together. 

unclear, please rephrase

throughout the results section use the past tense

Author Response

The manuscript has been modified according to the comments and has improved. Some details can be corrected as listed below.

We would like to thank the reviewers for their constructive remarks.

116 evacuation of the water surplus to the outside to avoid any risk of overflow (Figure d)

Fig 1 B

Answer: Thank you. This has been corrected accordingly.

119 maturation takes place

took place

Answer: Thank you. This has been corrected accordingly.

119 during the sampling period. Temperature rang between 24-30°C, pH, 7.2-8.1 and conductivity 1010-3115 µS.cm-1 120 .

please explain why these variations indicate “maturation”

Answer: We habe now specify that maturation indicates the process of colonization by microorganisms of the mud combined with thermal water.

were inoculated on solid medium with medium Z8 and Z8-salt [21]. Isolations were carried out 

maybe cite fig 1

Answer: Reference to Fig. 1 has been added, accordingly.

123 Samples from the mud itself and the basin walls collected the 07-24-2014 and the 08- 124 04-2014

Do these dates correspond to “mud maturation”? why were they chosen?

How long were these samples maintained and processed before the described analysis?

Answer: It has now been specified that mud was obviously mature at these specific dates.

Fig 1

please cite that the cyanobacteria shown in the table were explored in this study

  1. E) Liquid media culture of a monoclonal strain. 

of which genus?

  1. F) Microscopic view of the Dulcicalothrix strain PMC 884.14

not necessary

Answer: We have now specified that “Monoclonal strains could have been obtained for all cyanobacteria taxa except for Geitlernema we did success to maintain in culture.”

Genus of the strains shown in liquid culture was specified. We assume to illustrate the microscopic observation of the strain and to keep Fig. 1E.

155 Each cyanobacterial strains were

cyanobacterial strains were

Answer: This has been corrected accordingly.

169 DNA quality and quantity was checked with Qubit (Thermo)

only quantity, quality with spectrophotometer, Nanodrop or similar

Answer: Thank you. This has been modified accordingly.

164 DNA extraction from six biofilm mud samples from which cyanobacteria were 165 isolated and from the nine non-axenic cyanobacterial strains

Answer: This sentence has been corrected

189 gene was amplified from extracted DNA of the nine non-axenic cyanobacterial 190 cultures and five mud samples

6 or 5 mud samples? there are 6 in fig 5

Answer: Thank you. This has been corrected.

163 2.3. Genomic DNA extraction

described again in 187

Answer: Thank you. This has been modified in order to avoid such redundancy.

172 2.4. Molecular phylogeny of cyanobacterial strains

no need to describe this, already published in Duval et al [16]

Answer: This section has now been removed.

2.6. Metagenome sequencing and assembly of cyanobacteria’s strains microbial consortia

these subtitles can be confusing

please emphasize in 2.6 subtitle that this part refers to Metagenome-Assembled Genomes from cyanobacteria and heterotrophs

while in 2.5 it refers to 16S amplicon sequence to characterize the community composition or 16S rRNA metabarcoding sequencing

Answer: Thank you for this remark and these suggestions. Subtitles have now been modified accordingly.

252 also preserved

preserve

Answer: This has been corrected accordingly.

250-260 

appropriate to cite the time elapsed between collection and isolation of sample/strains and analysis 

Answer: This information has now been added within the M&M section “no later than 3 days after sampling”.

346-347

unclear statement

Answer: This sentence has been reworded.

373 and 379

and cyanobacterial strain cultures

Answer: This has been modified accordingly.

figure 4 C) only heterotrophs?

  1. D) ASV heatmap (<0.1%) ?

Answer: This information has been specified.

383-389 interestingly repeated

Answer: This has been corrected.

437 and 440 while metabarcoding 

Answer: Answer: This has been corrected.

Table 1 suggestion maintain a more detailed taxonomic affiliation, not just Phylum 

Answer: Ok, we have now added this information on table 1

  1. Differently

delete

Answer: This sentence has modified accordingly.

483-487

unclear statement

Answer: This sentence has now been clarified.

495 exploration of the cyanosphere

and cyanobacterial genomes

Answer: This sentence has modified accordingly.

510 Our results also support this assumption highlighting such potential metabolic 511 complementation within the cyanosphere

which assumption, that KEGG does not contain functions relevant to photosynthesis processes?

and why this  highlights potential metabolic complementation within the cyanosphere ?

Answer: Thank you. This paragraph has now been reworded accordingly.

514 According to their respectively more similar functional gene contents,?

Since they have similar gene contents (?)

Answer:

520 suggests that the different members of these clusters

within each cluster

Answer: This has been modified accordingly.

Fig 5  

Functional metagenomics of the cyanosphere from the nine cyanobacterial strains

Functional metagenomics of the cyanobacterial strains and their cyanosphere 

Answer: This has been modified accordingly.

  1. C) and D) of comparing cyanobacteria and heterotrophic bacteria metagenomes taken together. 

unclear, please rephrase

Answer: This has been modified.

throughout the results section use the past tense

Answer: When appropriate, past has been used through the result section.